# Effect of Flaxseed Gum on the Textural, Rheological, and Tribological Properties of Acid-Induced Soy Protein Isolate Gels

**DOI:** 10.3390/polym15132834

**Published:** 2023-06-27

**Authors:** Cunshe Chen, Peipei Ma, Siyuan Jiang, Imane Bourouis, Zhihua Pang, Xinqi Liu, Pengjie Wang

**Affiliations:** 1Key Laboratory of Geriatric Nutrition and Health, Ministry of Education, Beijing Technology and Business University, Beijing 100083, China; chencs@th.btbu.edu.cn (C.C.); mapeipei2021@163.com (P.M.); 18317776336@163.com (S.J.); bourouisiman701@gmail.com (I.B.); liuxinqi@btbu.edu.cn (X.L.); 2Department of Nutrition and Health, China Agricultural University, Beijing 100083, China; wpj1019@cau.edu.cn

**Keywords:** soy protein gel, flaxseed gum, texture, rheology, tribology

## Abstract

This study aimed to investigate the effects of incorporating different concentrations of flaxseed gum (FG) into acid-induced soy protein isolate (SPI) gels. The investigation focused on assessing the effects of FG on the textural, rheological, and tribological properties of the resultant SPI gels. The results showed that adding a small amount of FG (0.05%) to the SPI gel system increased the storage modulus (G′) and enhanced gelation while improving textural properties including hardness, viscosity, elasticity, and adhesion. Moreover, these gels exhibited strong water-holding capacity, a desirable property in various food products. However, when the concentration was increased to 0.3%, the WHC of the gel decreased, as did the hardness and cohesiveness. The particle size of the gel also increased with increasing concentration. Tribological investigations revealed that at 0.05–0.2% FG addition, the coefficient of friction (μ) of the composite gel was decreased compared to the pure SPI gel. In the sliding speed range of 1–100 mm/s, the coefficient of friction gradually increased with increasing concentration. When the FG concentration was 0.05%, the μ of the gel system was the lowest. In summary, low concentration of FG (0.05%) was found to play an important role in improving the properties of SPI gel, including enhancing textural, rheological, and lubricating properties.

## 1. Introduction

Hydrogels are polymeric materials with a three-dimensional network structure using water as a dispersion medium and can be formed by physical or covalent cross-linking of polymers [1]. They are ideal for a variety of food products due to their excellent textural and rheological properties [2]. Food gels have excellent potential to improve food texture, promote innovative food products, and enhance bioavailability and nutritional fortification. Proteins are a high-quality food resource with high nutritional value and rich functional properties. Among them, plant proteins with good biocompatibility, safety, and sustainability are favored by many scientists and can be used as fat substitutes, for bioactive substance encapsulation and release, as emulsion stabilizers and texture modifiers, and so on [3,4,5]. Soy protein isolate (SPI) is widely recognized as a high-quality protein source with well-balanced amino acids and exceptional functional and processing properties [6,7]. Previous studies have shown that nanogel prepared with SPI and sodium alginate can be combined with cress seed gel to obtain a more elastic and dense composite gel system, which can be used as a curcumin delivery system [8].

In particular, SPI has received considerable attention in the food industry due to its gelling properties, which enhance the texture of food products [7]. One of the effective ways to prepare SPI gels involves the acidification of heated SPI with glucono-δ-lactone (GDL) to create a three-dimensional network structure [9]. Soy protein has an isoelectric point of 4.5, rendering it negatively charged under neutral conditions. Additionally, soy protein undergoes denaturation during heat treatment, producing a soluble complex between protein molecules [10]. GDL slowly releases protons (H+) to neutralize the negative surface charge of denatured soy protein, weakening the electrostatic repulsion between the protein molecules, enabling their aggregation through hydrophobic interactions and leading to gel formation [11]. The formation of SPI gels can be influenced by a number of factors, and the addition of different ingredients can have different effects on the production of the concerned product.

It was reported that incorporating polysaccharides and dietary fibers improves the functional properties of food products [12]. These compounds have been shown to bind to proteins and form multiple network structures via hydrogen bonds, van der Waals forces, and hydrophobic interactions, which play a critical role in improving the functional and physicochemical properties of protein gels [13].

Various studies have investigated the effect of different sugars and natural edible gums on the rheological properties of acid-induced SPI gel hybrids [14,15,16,17,18]. Among them, FG is known as a heteropolysaccharide comprising neutral and acidic polysaccharides [19,20] and has been found to possess potential health benefits, such as preventing coronary heart disease, diabetes, and colon cancer, and reducing obesity [21,22]. With its health benefits and high nutritional value, FG has been included in the pharmacopeia of the United States and other countries and has potential value in the food industry. Bi et al. [23] reported the effect of FG on the rheological properties, microstructure, and gel properties of soy protein isolate (SPI) acid-induced emulsion gels, showing that the appropriate concentration of FG has the ability to improve the water retention and stability of SPI emulsion gels. Chen et al. [24] have found that the FG can enhance the consistency of peanut isolate and enable it to form a more elastic gel in a shorter period of time. Most of the previous studies have focused on the effect of FG on the rheological properties of SPI gel. However, the impact of flaxseed gum (FG) on the textural and tribological properties of the GDL-induced SPI gel system has not yet been explored.

The physical properties of food products play a crucial role in consumer evaluation. Textural properties, including viscosity, water-holding capacity, and hardness, significantly influence both the processing performance and sensory attributes of the products [25]. Furthermore, relative surface movements occur between food particles and oral surfaces during the consumption of food. The resulting oral frictional resistance directly affects sensory properties such as smoothness and creaminess [26]. Therefore, developing SPI-based gels that exhibit superior textural, rheological, and lubricating properties is crucial for product acceptance. The purpose of this study is to provide insights into the effect of FG on the formation of GDL-induced SPI gels and their physical properties, including the gel strength and textural and tribological properties, to improve the quality of soy-protein-based food products.

## 2. Materials and Methods

### 2.1. Materials

SPI was purchased from Linyi Shan Song Biological Products Co., Ltd. (Linyi, China), with a protein content of 93.2%. Flaxseeds were purchased from the local market(Beijing, China). Ethanol (95%) was purchased from Fujian Chemical Reagent Co. (Fujian, China). GDL was obtained from Xingzhou Medicine Food Co., Ltd. (Anhui, China). Polydimethylsiloxane (PDMS, CAS: 9006-65-9) was obtained from Dow Corning Deland, America). All other chemicals used were analytical grade.

### 2.2. Sample Preparation

#### 2.2.1. SPI Solution Preparation

The SPI solution preparation was carried out according to the literature [27]. The SPI powder was weighed and dissolved in deionized water to prepare SPI stock solutions with 9% and 12% mass ratios. The solutions were stirred at 500 rpm for 6 h at 25× and then hydrated overnight in a refrigerator at 4 °C. The hydrated SPI stock solutions were centrifuged(AllegraX-15R, Beckman Coulter, Beijing, China) at 233× *g* for 10 min, and the filter residue was carefully separated and set aside.

#### 2.2.2. FG Extraction

FG was extracted using hot water extraction in accordance with Kaushik et al. [28], with little changes. Briefly, the flaxseed was soaked in Milli-Q water at a flaxseed-to-water ratio of 1:18 (*w/w*) at 90 °C with continuous and gentle stirring over a magnetic plate for 2 h for each of two consecutive cycles of extraction. After extraction, FG was precipitated using three volumes of 95% ethanol. The precipitated gum was collected by centrifuging at 4000× *g* for 10 min. The resulting gum was then concentrated in a blast dryer set at 50 °C. Once concentrated, it was freeze-dried and ready for use. The particle size of FG obtained by the above method is between 200 and 500 nm [29].

#### 2.2.3. SPI-FG Gels Preparation

The SPI-FG gel preparation was carried out according to the literature [30], with some modification. The SPI solution was formulated to a protein concentration of 5% based on the dissolved protein content. FG was added at concentrations of 0.05%, 0.1%, 0.2%, and 0.3% relative to the total solution volume. The mixed composite system was placed in a water bath magnetic stirrer at 100 °C for 20 min, and then the samples were removed from the water bath and quickly cooled down to 40 °C in an ice water bath. Subsequently, GDL was added to the dispersions at 1.5% concentration and incubated in a water bath at 40 °C for 4 h. After incubation, the samples were cooled down to 10 °C and then refrigerated in a 4 °C biochemical incubator for 48 h.

### 2.3. Texture Analysis and Water-Holding Capacity (WHC)

Texture analysis of the stored samples was carried out using a CT3 texture analyzer (Brookfield, Middleboro, MA, USA) and following the method of Lau et al. [31]. A cylindrical probe of 12.7 mm diameter was used at a temperature of 4 °C and a test speed of 0.5 mm/s; the test distance was set at 5 mm, and the trigger force at 5 g. Hardness, elasticity, viscosity, and cohesion were recorded. The hardness was defined as the maximal force applied (g); viscosity was defined as the area of the first negative peak (mJ); elasticity was defined as the recovery height of the sample detected in the second compression (mm); and cohesion was the area of work during the second compression divided by the area of work during the first compression.

To determine the water-holding capacity (WHC), stored samples were centrifuged at 233× *g* for 10 min at 4 °C. The WHC was calculated using the following equation, where m1 is serum expelled weight (g), and m2 is SPI-FG gel weight, which is 10 g [32]:WHC = (m1/m2) × 100%(1)

### 2.4. Particle Size

The particle size distribution of gel samples was analyzed using a laser diffraction particle size analyzer (SALD-2300, Shimadzu Corporation, Tokyo, Japan) according to the method reported previously [33]. The gel samples, stored at 4 °C for 48 h, were dispersed 10-fold with distilled water and stirred well before analysis. The refractive index of the particles was set at 1.45 with a precision of 0.001. The particle size polydispersity was characterized by the size distribution span using the equation: span = (D90 − D10)/D50, where D90, D10, and D50 are the volume diameters at 90%, 10%, and 50% of the cumulative size, respectively. Three replicates of each sample were tested.

### 2.5. Tribology

The lubricity properties of the gel samples were measured using a plate friction rheometer (TA Instruments, Newcastle, NSW, USA) and PDMS (surface roughness < 50 nm) simulated oral cavities. The PDMS surface was prepared by mixing the cross-linker and base fluid with a ratio of 1:10, as described in the product specification, and removing the entrapped air by vacuuming. The resulting liquid is cast using a smooth stainless-steel mold and cured overnight in an oven at 70 °C. The experiments aimed to assess the frictional characteristics of the samples during simulated oral processing. A constant normal force of 2 N, corresponding to a pressure of 27.83 KPa, was applied to represent the moderate normal force exerted on the sample during oral processing. For each test, 3 g of gel sample was carefully applied, covering the lower plate surface at a thickness of approximately 2 mm. The friction results between the tribo-rheometer and the tape surface were recorded at speeds ranging from 0.01 to 100 s^−1^. Two replicates of each gel sample were tested, with each replicate consisting of three parallel measurements. The tribological data were screened to remove data points with errors greater than 5% in the analysis results.

### 2.6. Viscoelasticity

The viscoelastic properties of the gel samples were measured using the same rheometer used for tribology analysis, equipped with a 40 mm diameter parallel plate. A gap size of 1 mm was used. The measurement was conducted by increasing the frequency from 0.01 to 10 Hz at 37 °C. A strain of 0.5% was applied (within the linear viscoelastic range). The storage modulus (G′), loss modulus (G″), and complex viscosity (η*) were recorded. The loss tangent is derived from the equation: tan δ = G′/G″ [34].

### 2.7. Data Processing and Analysis

Data were analyzed using SPSS software 26.0 (SPSS, Chicago, IL, USA). All the experiments were performed in triplicate, and the results are expressed as the mean of independent experiments with standard deviation. Experimental data were analyzed by one-way ANOVA (pairwise comparison of means with Tukey HSD post hoc test) to determine the differences in samples.

## 3. Results

### 3.1. Viscoelastic Properties

To evaluate the viscoelasticity of the composite gel system with different concentrations of flaxseed gum (FG), storage modulus (G′) and loss modulus (G″) were measured at frequencies ranging from 0.1 to 10 Hz. G′ represents the ability of a viscoelastic material to store energy in one cycle under alternating stress, typically reflecting the variation in the elasticity of the sample. Conversely, G″, which is the energy lost when a material is irreversibly deformed due to viscosity, reflects the magnitude of the material’s viscosity [35]. A smaller tanδ and a higher complex viscosity implies a more homogeneous network structure and stronger gel properties [36]. As illustrated in Figure 1, all the samples showed similar shapes and trends in terms of elastic (G′) and viscous (G″) moduli, which increased with increasing frequency. This may be related to the change in the structure of the material due to small deformation mechanical work, which leads to a decrease in the size of the average clusters and a rapid reorganization, forming more connections [37]. At the same time, the composite viscosity also gradually decreased with the increase of frequency. The frequency dependence of the G′ and G″ values were in good agreement with previous results observed for Mesona blumes polysaccharide–soy protein isolates mixed gel [38] and various types of hydrocolloids–soy protein gel [39]. G′ exhibited higher values than G″ values over the entire frequency range, indicating that all SPI gels, with or without FG, showed viscoelastic character and the elastic component of the samples dominated over the viscous component (G″) throughout the frequency range [40]. At a frequency of 1 Hz, the tanδ of the samples is between 0.19 and 0.27 (Table 1), which indicates that these samples can form weak gel networks with solid-like behavior [8]. Adding 0.05% FG to the gels increased the G′ of the gels significantly and decreased the tanδ, indicating that the addition of FG at this concentration promoted the formation of cross-links between macromolecules and a stronger gel network structure [18]. This conclusion is also illustrated by the fact that the maximum loss tangent can be observed in this gel. Thus, the elasticity of the SPI-FG gels was enhanced at this concentration [41]. However, the G′ and G″ of the gels were significantly lower at 0.1% and 0.2% flaxseed gum concentrations, possibly due to the phase separation process during gelation. Further increasing the FG concentration (0.3%), the sample showed comparable G′ and η* values to that of 0.05% FG-containing gel. The phenomenon might be related to excess FG, which did not interact or react with SPI. A study by He et al. [42] found that low concentrations of pectin promoted the texture and gelability of ginkgo seed protein isolate (GSPI)–pectin composite gels, while high concentrations of pectin can lead to overfilling that adversely affects the gel structure. These FG aggregated themselves. The enlarging volume fraction of the FG phase disrupted the continuous SPI gel network and further disordered the SPI gel structure, resulting in phase separation. The phase separation and disordered structure of the gel led to higher modulus values (G′ and G″) with a high concentration of FG [43]. Similar results have been reported in SPI gels with locust bean gum. A high concentration of polysaccharides could have resulted in a highly compacted structure, revealed by a high G′ [44].

### 3.2. Tribology

The viscoelastic properties of a gel can significantly impact its lubricating properties, which can be measured by analyzing the variation in friction using the Stribeck curve model. The curve is divided into three zones, namely the boundary zone, the mixed zone, and the hydrodynamic zone, each corresponding to a different fluid entrainment mechanism between the two contact surfaces [45]. Figure 2 shows the Stribeck curves of the five samples, with the boundary, mixed, and hydrodynamic zones clearly distinguishable.

At low sliding speed (between 0 and 1 mm/s), the samples are in the boundary zone, where the friction coefficient remains relatively constant and little difference is observed between the different FG concentrations.

This can be attributed to the formation of a lubricating film as the particles in the gel adsorb on the PDMS, resulting in a low coefficient of friction [27,46]. The constant friction coefficient then started transiting from the boundary regime to the mixed regime, which could be allied with the particle entrainment facilitated by the sample’s viscosity [47]. In this regime, surface properties, particle characteristics, and sample viscosity determine the friction coefficient. The friction is no longer a decreasing value like the conventional Stribeck curve but increases linearly with speed. Since SPI gel is a soft gel structure, its lubrication behavior is very similar to fluid gel. Therefore, the shape of the friction curve observed in this work is very similar to that of the agarose gel solution reported by [48], except in the boundary zone, which is constant in this work. It can be seen from the results that the friction curve of gel samples incorporated with 0.05–0.2% FG showed a lower friction coefficient than the control during this regime. This is most likely due to the FG acting as a lubricant between the protein particles and the contact surface, resulting in a lower friction coefficient. Previous studies have reported that certain polysaccharides enhanced lubrication by adsorption, yielding a hydrated, viscoelastic film [33]. However, it is noteworthy that the effect of excessively high colloidal content on the lubrication of the system is more pronounced, as indicated by the increased friction coefficient with increasing FG concentration, where the friction coefficient increases in the order of 0.05%, 0.1%, and 0.2% FG. This is likely due to the high colloid content causing excessive aggregation of the protein particles, which prevents the relative displacement of the contact surface and increases the resistance. The effect of high colloid content may be associated with the increasing average particle size, which increases with the increase of FG concentration (Table 2). At higher sliding speed, the mixed system enters the hydrodynamic zone, where the friction of the solution is mainly related to the viscosity of the sample [49], resulting in a slight overall downward trend in the friction coefficient.

### 3.3. Particle Size Distribution

Particle size is an important factor that can significantly affect the rheological properties of gels [30]. Smaller particle size can enhance the interaction between protein molecules and the added components, resulting in a more homogeneous gel network, which has the potential to improve gel strength [50].

From Table 2, it can be seen that the addition of FG reduced the span value of gel particles to between 1.5% and 2.5%, and the particle size distribution became more uniform. The average particle size increased with increasing FG concentration. The gels with a 0.3% FG addition exhibited the largest particle size. It is possible that the addition of flaxseed gum leads to volume repulsion between protein molecules, resulting in depletion interactions [51]. Solutions with larger particle diameters would lead to gels with larger aggregates and lower WHC. Similar results have been shown by Xia et al. [52], who found that the pore sizes of SPI gel were larger when the soybean soluble polysaccharide (SSPS) addition was higher. The increasing FG addition led to a larger particle size in the SPI-FG mixture, with larger aggregates causing less hydrophobic regions to be exposed on the surface. With increasing FG concentration, fewer SPI aggregates were formed due to there being fewer hydrophobic regions, resulting in gels with lower gel strength and decreased water-holding capacity (WHC) [52]. Wang et al. [41] also reported that adding Mesona blumes polysaccharide increased the particle size of GDL-induced SPI gels. However, Pang et al. [44] demonstrated that adding 0.01 and 0.05% guar gum and locust bean gum (LBG) reduced the D50 of soymilk gel, while adding 0.1% gum increased the values. This suggests that different polysaccharides may have varying effects on particle size distribution depending on their properties and interactions with the gel matrix.

### 3.4. Texture Analysis and Water-Holding Capacity (WHC)

Texture is an essential factor that impacts the sensory perception and consumer acceptance of food products. WHC indicates the gel’s ability to retain water. Table 3 presents the textural properties and WHC of the composite gel system at different concentrations. WHC is influenced by the microstructure and inter-particle bonding within the gel matrix, and a higher water-holding capacity indicates a stronger bonding between particles and enhanced water retention capacity [53]. In this study, adding FG at concentrations of 0.05% to 0.2% did not significantly increase WHC compared to the control gel. However, at a concentration of 0.3%, a significant decrease in WHC was observed. This decrease could be attributed to the protein particles gathering around the FG, leading to larger pore sizes between adjacent particles and a less dense mesh structure. Additionally, the involvement of less soy protein in the gel formation at this concentration might contribute to weaker gels with reduced water-holding capacity [52].

The hardness of the gel tended to decrease as the concentration increased. Similar results were obtained by Pang et al. [44]; adding guar gum decreased the firmness of soymilk gel at concentrations of 0.01 and 0.1% and decreased WHC at 0.01–0.1% concentrations, suggesting that it should be due to the depletion flocculation. Ingrassia et al. [35] also reported that the firmness of gels decreased significantly with increasing concentrations of tara gum (TG) in SPI/TG gels. These observations indicate that the presence of TG affects the protein gelation process, leading to the formation of weaker gels, suggesting that this behavior is linked to the competition between SPI acid gelation and phase separation processes caused by the thermodynamic incompatibility between SPI and TG. However, at a concentration of 0.05%, an increase in the hardness, viscosity, and elasticity of the gels was observed. This is most likely because the low concentration of linseed gum can interact with SPI, and the gel properties are enhanced by the filling of the gum. This suggests that a small amount of FG can enhance the interaction between protein particles, leading to a denser structure and increased viscosity and elasticity [23]. These findings correlated well with our rheology results, where adding 0.05% FG increased the G′ of the gels significantly. Cohesiveness indicates the resistance to breaking down of the gel’s internal structure [54]. It increased significantly with increasing FG concentration, then decreased at 0.3% FG.

## 4. Conclusions

This study successfully investigated the effects of flaxseed gum (FG) on GDL-induced SPI gel systems, aiming to enhance the textural and tribological properties of the gels. Through viscoelastic analysis, it was found that adding FG influenced the viscoelastic properties and gel network structure of the SPI gels. Specifically, an optimal concentration of FG (0.05%) significantly enhanced the gel elasticity, resulting in improved qualitative properties such as hardness, elasticity, and viscosity. However, higher concentrations of FG led to phase separation and disordered gel structures, which resulted in reduced moduli. Moreover, the incorporation of FG at different concentrations revealed variations in the textural properties of the gels. Notably, adding 0.05% FG increased the values of hardness, elasticity, and viscosity. Furthermore, including FG effectively reduced the friction coefficient of the SPI gel, which exhibited excellent lubricity at the 0.05% concentration. This can be attributed to the promotion of a stronger gel network structure and the lubricating effect of FG between protein particles and the contact surface. These findings highlight the potential of FG as a valuable additive for enhancing the gelation properties of SPI while introducing desirable textural, rheological, and lubricating effects. The knowledge gained from this study can find practical applications in various food industry sectors where gel properties play a crucial role in product formulation, stability, and consumer acceptance. The achievement of our research aims and the originality of this work lie in the identification of the optimal concentration of FG that enhances gel properties and improves textural properties, and the novel exploration of FG as a lubricating agent in SPI gels.

## Figures and Tables

**Figure 1 polymers-15-02834-f001:**
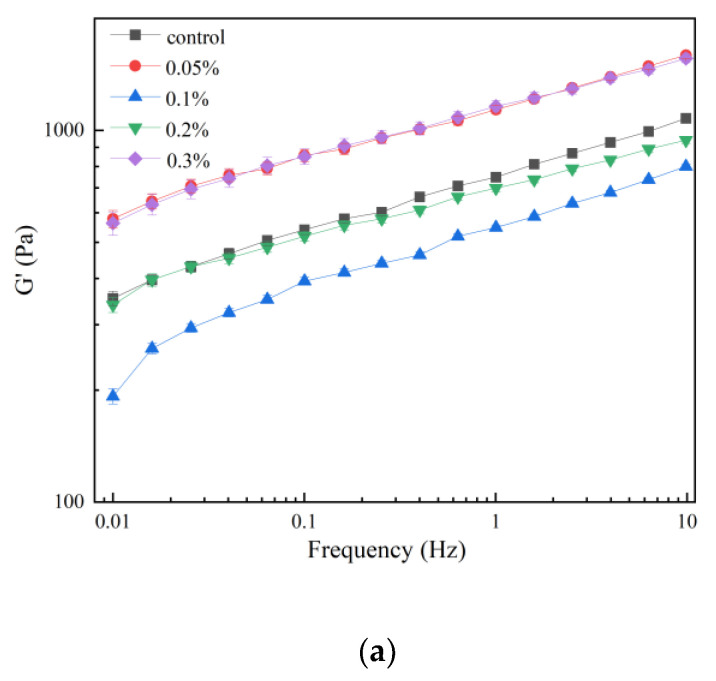
The (**a**) storage modulus (G′), (**b**) loss modulus (G″), and (**c**) loss tangent (tanδ) of SPI/linseed gum hybrid gels at different linseed gum concentrations.

**Figure 2 polymers-15-02834-f002:**
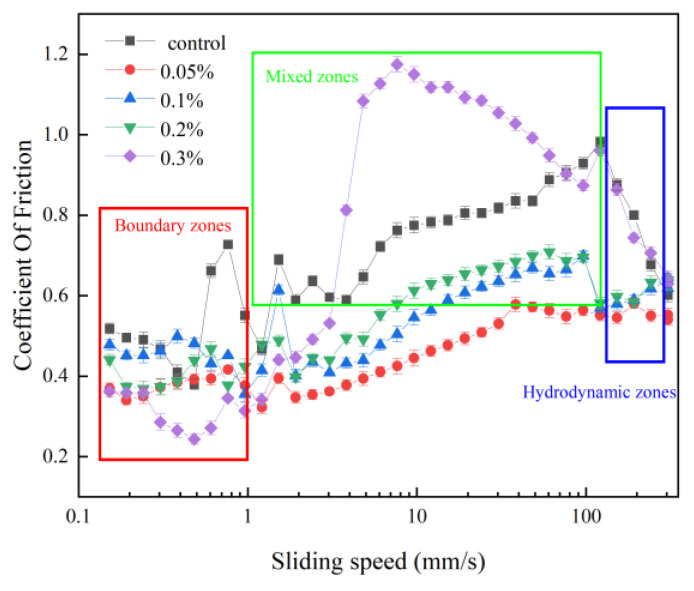
Friction curves of SPI/FG mixed gels at different FG gradient concentrations.

**Table 1 polymers-15-02834-t001:** Rheological parameters of SPI/linseed gum hybrid gels at different linseed gum concentrations determined by frequency sweep test at f = 1 Hz, constant strain of 0.5%, and 37 °C.

Sample	G′ (Pa)	G″ (Pa)	tan δ	η *
Control	966.22	259.74	0.269	181.98
0.05%	1487.82	276.95	0.186	243.27
0.1%	697.84	158.05	0.226	115.38
0.2%	898.07	199.51	0.222	146.42
0.3%	1493.42	284.52	0.191	244.88

**Table 2 polymers-15-02834-t002:** Differences in the particle size of gels with different FG concentrations.

100 °C	Average Particle Size (μm)	D10	D50	D90	Span
Control	3.54 ± 0.45 ^a^	0.27 ± 0.08 ^a^	9.85 ± 0.47 ^a^	43.72 ± 0.88 ^a^	4.41 ± 0.38 ^b^
0.05%	21.43 ± 0.29 ^b^	10.23 ± 1.00 ^b^	22.85 ± 0.96 ^b^	49.05 ± 5.07 ^b^	1.70 ± 0.13 ^a^
0.1%	22.72 ± 0.72 ^c^	10.78 ± 2.44 ^b^	22.90 ± 0.42 ^b^	61.99 ± 4.72 ^c^	2.24 ± 0.33 ^a^
0.2%	40.94 ± 3.26 ^d^	17.47 ± 0.37 ^c^	43.17 ± 0.39 ^c^	94.19 ± 2.33 ^d^	1.78 ± 0.40 ^a^
0.3%	49.90 ± 1.75 ^e^	20.59 ± 0.44 ^d^	56.97 ± 1.34 ^d^	102.88 ± 1.80 ^e^	1.44 ± 0.19 ^a^

Note: Differences between data in each column containing the same letter are not significant (*p* > 0.05).

**Table 3 polymers-15-02834-t003:** Textural analysis and water-holding capacity (WHC) of gels at different concentrations.

100 °C Treatment	Water Holding Capacity (%)	Hardness (g)	Viscosity (mJ)	Elasticity (mm)	Cohesiveness
Control	91.20 ± 3.21 ^bc^	40.65 ± 0.77 ^a^	0.09 ± 0.01 ^b^	3.73 ± 0.04 ^d^	0.30 ± 0.01 ^a^
0.05%	90.42 ± 0.47 ^b^	41.45 ± 0.49 ^a^	0.24 ± 0.04 ^d^	4.04 ± 0.00 ^e^	0.37 ± 0.04 ^c^
0.1%	92.71 ± 1.58 ^c^	14.15 ± 0.60 ^b^	0.02 ± 0.01 ^a^	1.94 ± 0.16 ^b^	0.43 ± 0.07 ^d^
0.2%	91.60 ± 3.22 ^bc^	12.15 ± 0.67 ^b^	0.08 ± 0.01 ^b^	2.87 ± 0.31 ^c^	0.55 ± 0.042 ^e^
0.3%	82.40 ± 2.79 ^a^	11.45 ± 0.40 ^b^	0.10 ± 0.01 ^c^	2.87 ± 0.31 ^c^	0.32 ± 0.045 ^b^

Note: Differences between data in each column containing the same letter are not significant (*p* > 0.05).

## Data Availability

The data presented in this study are available on request from the corresponding author.

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
