# Peer review of "Effect of Flaxseed Gum on the Textural, Rheological, and Tribological Properties of Acid-Induced Soy Protein Isolate Gels"

_polymers, 2023, doi:10.3390/polym15132834_

Round 1

Reviewer 1 Report

General comments: The study investigated the effect of flaxseed gum on textural, rheological, and tribological properties of GDL-induced soy protein gels. The topic is within the scope of the journal, and the study design is appropriate. However, I have a concern regarding the particle size analysis of the gel samples (see my specific comments). I also believe analyzing the microstructure of the gel samples could further strengthen the study. Furthermore, there are several grammatical and syntax errors. The authors should thoroughly check for such errors or refer the manuscript to a native English speaker to revise it. Additional queries, comments, and suggestions can be found in my specific comments.

Specific comments

Lines 21-22: Change “a wide range of” to “well balanced”.

Line 26: This is only one way of preparing SPI gels, not the standard method.

Line 29: “However” should be used to contradict the previous statement, which is not the case here.

Line 36: “van der Waals”.

Lines 51-53: Run-on sentence. Split it into two sentences.

Lines 65: Provide the relative centrifugal force instead of rpm.

Lines 67-75: Provide more details for this section.

Line 89: Define these parameters.

Line 96: Did the authors mean “sol samples”. How can they dilute a gel sample?

Line 104: What does “PDMS” stand for?

Line 123: Which post-hoc test was performed?

Lines 133-134: Awkward sentence. Please rephrase.

Figure 1: Use two separate plots for G’ and G”. The current figure has too many overlapping symbols.

Line 169: Adsorb to what? Confusing sentence.

Figure 2: Consider label the three different zones in the figure.

Lines 196-203: Provide a reference.

Line 208: What does “SSPS” stand for?

Line 214: What does “LBG” stand for?

Line 215: The number of the first D value is missing.

Line 216: What is the particle size of the flax gum? The authors should provide the data.

Table 1: How did the authors measure the particle size of the gel samples? The laser diffraction particle size analyzer only measures liquid or powder samples.

Line 219: Delete “in the same corner”.

Lines 231-233: The authors should run an SEM to validate this assumption.

Lines 245-251: Similar results were observed when adding pectin to ginkgo seed protein gels. Incorporation of low concentrations of pectins enhanced various textural properties while high concentration of pectins weakened the gel texture. Such polysaccharides are filler ingredients for protein gels. Overfilling of excessive polysaccharides can have detrimental effect on the gel structure. Reference: He, Z., Liu, C., Zhao, J., Li, W., & Wang, Y. (2021). Physicochemical properties of a ginkgo seed protein-pectin composite gel. Food Hydrocolloids, 118, 106781.

Line 251: Provide an explanation.

Line 255: Delete “in the same corner”.

There are several grammatical and syntax errors. The authors should thoroughly check for such errors or refer the manuscript to a native English speaker to revise it. 

Reviewer 2 Report

The manuscript examined the Effect of flaxseed gum on the textural, rheological, and tribological properties of acid-induced soy protein isolate gels. This article does not have any particular novelty and aim of investigation is not clear. I recommend it for publication after consideration of the following comments.

Title: Well-deserved

Abstract:

It is not well stated with details and results

Keywords: Related selected

Introduction:

The introduction was written very briefly; please expand on it

·        The Introduction section of the paper should give greater context to the project. You could write more expressively and clearly and used more relevant investigation https://doi.org/10.1186/s40538-022-00304-4

English should be thoroughly revised. Grammatical and vocabulary errors appear with a specific frequency. Some sentences need to be understood.

Materials and Methods:

Please write materials as Company Name (City, Country), especially for the chemical analysis assessment used in the study.

Please add the references of methods.

Results and Discussion

As shown by this study, you have gained a general understanding of the effects of incorporating different concentrations of flaxseed gum into acid-induced soy protein isolate (SPI) gels.

The friction coefficient results Interpret with the adhesion results in such a way that the data continuity is coherent.

Complex viscosity and tangent delta have not been investigated.

Conclusions:

Conclusions are far too general and vague. These need to be precise and explain how the aims of the paper have been achieved while emphasizing the originality of the work.

References:

They were completely relevant

-

Round 2

Reviewer 1 Report

The authors have addressed all my comments and revised the manuscript accordingly. The new figures seem to have very poor resolution. Please make sure to use high resolution figures.

NA

Reviewer 2 Report

The authors covered all the problems I found to my previous review.

The manuscript can be accepted in the present version.Only, the figures quality should be improved.

-